

# Fossil fern rhizomes as a model system for exploring epiphyte community structure across geologic time: evidence from Patagonia

Alexander C. Bippus[1], Ignacio H. Escapa[2], Peter Wilf[3] and Alexandru M.F. Tomescu[4]

[1] Department of Botany and Plant Pathology, Oregon State University, Corvallis, OR, United States of America
[2] Consejo Nacional de Investigaciones Cientificas y Tecnicas (CONICET), Museo Paleontológico Egidio Feruglio, Trelew, Argentina
[3] Department of Geosciences, Pennsylvania State University, University Park, PA, United States of America
[4] Department of Biological Sciences, Humboldt State University, Arcata, CA, United States of America

Corresponding author
Alexander C. Bippus,
acb613@humboldt.edu,
bippusa@oregonstate.edu

## ABSTRACT

**Background**. In extant ecosystems, complex networks of ecological interactions between organisms can be readily studied. In contrast, understanding of such interactions in ecosystems of the geologic past is incomplete. Specifically, in past terrestrial ecosystems we know comparatively little about plant biotic interactions besides saprotrophy, herbivory, mycorrhizal associations, and oviposition. Due to taphonomic biases, epiphyte communities are particularly rare in the plant-fossil record, despite their prominence in modern ecosystems. Accordingly, little is known about how terrestrial epiphyte communities have changed across geologic time. Here, we describe a tiny *in situ* fossil epiphyte community that sheds light on plant-animal and plant-plant interactions more than 50 million years ago.

**Methods**. A single silicified *Todea* (Osmundaceae) rhizome from a new locality of the early Eocene (ca. 52 Ma) Tufolitas Laguna del Hunco (Patagonia, Argentina) was studied in serial thin sections using light microscopy. The community of organisms colonizing the tissues of the rhizome was characterized by identifying the organisms and mapping and quantifying their distribution. A $200 \times 200\ \mu\text{m}$ grid was superimposed onto the rhizome cross section, and the colonizers present at each node of the grid were tallied.

**Results**. Preserved *in situ*, this community offers a rare window onto aspects of ancient ecosystems usually lost to time and taphonomic processes. The community is surprisingly diverse and includes the first fossilized leafy liverworts in South America, also marking the only fossil record of leafy bryophyte epiphytes outside of amber deposits; as well as several types of fungal hyphae and spores; microsclerotia with possible affinities in several ascomycete families; and evidence for oribatid mites.

**Discussion**. The community associated with the Patagonian rhizome enriches our understanding of terrestrial epiphyte communities in the distant past and adds to a growing body of literature on osmundaceous rhizomes as important hosts for component communities in ancient ecosystems, just as they are today. Because osmundaceous rhizomes represent an ecological niche that has remained virtually unchanged over time and space and are abundant in the fossil record, they provide

a paleoecological model system that could be used to explore epiphyte community structure through time.

## INTRODUCTION

In the modern biota, direct access to living organisms has revealed significant portions of their networks of ecological interactions. In contrast, understanding of such interactions is vastly incomplete in ecosystems of the geologic past. Here, we investigate a complex community of organisms that lived in association with an osmundaceous fern, preserved in Eocene rocks (ca. 52 Ma) of the Huitrera Formation, Argentinean Patagonia. The Huitrera Formation hosts one of the most diverse Eocene floras characterized to date (*Wilf et al., 2003*; *Wilf et al., 2005a*), yet the interactions of plants in this flora with other organisms are just beginning to be characterized (e.g., *Wilf et al., 2005b*). The fossil community associated with the osmundaceous fern is surprisingly diverse and includes the first fossil epiphytic liverworts known outside of amber deposits. Preserved *in situ*, this community offers a rare window onto aspects of ancient ecosystems usually lost to time and taphonomic processes. Such ecologically relevant aspects of community dynamics (abundance and diversity of epiphytes and incidence of herbivory), typically unavailable for fossil associations, are described here. Rather than describing in detail the biotic interactions between members of this community, here we aim to provide an assessment of epiphyte community structure. We also use a broad definition of epiphyte, which encompasses all organisms that live on a host plant (*Steel & Bastow Wilson, 2003*). Together, the fossils represent a tiny epiphyte community centered around a single host plant.

## MATERIALS AND METHODS

### Geologic setting

The fossils described here come from a new site ca. 2.1 km south of the Chubut River and 9.8 km due west of Piedra Parada in northwestern Chubut Province, Patagonian Argentina, S 42°39′20.60″, W 70°13′22.20″. The source strata belong to the Tufolitas Laguna del Hunco, an early Eocene, volcaniclastic, lacustrine caldera-fill deposit of the Huitrera Formation that is embedded in the Middle Chubut River Volcanic-Pyroclastic Complex (*Aragón & Mazzoni, 1997*). The new fossil site sits in the southern exposure area of the Tufolitas Laguna del Hunco, ca. 24 km SSW of the well-known Laguna del Hunco fossil locality (*Berry, 1925*; *Petersen, 1946*; *Wilf et al., 2003*), which lies in the northeastern exposures of the same extensive, highly fossiliferous unit. In this southern area, an ignimbrite that caps the fossil lake bed exposures ca. 5 km east of the new fossil site yielded an $^{40}$Ar-$^{39}$Ar age on plagioclase of 49.26 $\pm$ 0.56 Ma (early Eocene, Ypresian) in an unpublished thesis (*Gosses, 2006*; *Gosses et al., 2006*), directly providing a minimum age for the osmundaceous

rhizome. We note that this age has not been subsequently vetted or revised for updated decay constants (*Kuiper et al., 2008*), but it is likely to be broadly accurate because it lies in correct stratigraphic order relative to other dated samples from the Eocene caldera system (*Gosses, 2006*; *Gosses et al., 2006*; *Tejedor et al., 2009*).

At Laguna del Hunco (northeastern exposures), three $^{40}$Ar-$^{39}$Ar ages from volcanic ashes and two recorded paleomagnetic reversals, all from strata located within the main fossiliferous section of the Tufolitas Laguna del Hunco, constrain the time of fossil deposition there to the early Eocene as well (*Wilf et al., 2003*; *Wilf et al., 2005a*). The most reliable radiometric age at Laguna del Hunco, on sanidines from an ash taken from the middle of the most densely sampled fossiliferous interval and analyzed in two different labs, is 52.22 ± 0.22 Ma (early Eocene, Ypresian) following recalibration for modern decay constants (*Wilf et al., 2003*; *Wilf et al., 2005a*; *Wilf et al., 2017*; *Kuiper et al., 2008*; *Wilf, 2012*). Thus, the osmundaceous rhizome can safely be considered Ypresian and in the age range of ca. 49.3 to ca. 52.2 Ma. We prefer the older end of this range because the dates from Laguna del Hunco, despite their ca. 24 km geographic distance from the new fossil site, come from tuffs located within the Tufolitas Laguna del Hunco, the same unit that holds the osmundaceous rhizome fossil. The dated ignimbrite in the southern area (*Gosses, 2006*; *Gosses et al., 2006*) lies above that unit.

At Laguna del Hunco itself, the Tufolitas Laguna del Hunco host one of the most diverse Eocene compression floras known (*Wilf et al., 2003*; *Wilf et al., 2005a*). The flora has a robust Gondwanic component displayed among its gymnosperm and angiosperm species, as described extensively elsewhere; among the most striking occurrences are the well-preserved and abundant fossils of *Agathis* (Araucariaceae) and *Eucalyptus* (Myrtaceae) (*Gandolfo et al., 2011*; *Wilf et al., 2013*; *Wilf et al., 2014*).

### The host fern

The host plant is a permineralized osmundaceous fern rhizome segment ca. 8 cm in diameter and 20 cm tall. This rhizome specimen and associated sterile foliage collected from the same locality were recently described as *Todea* cf. *T. amissa* M. Carvalho (*Bomfleur & Escapa, 2019*), a species initially described based on compressions of sterile and fertile foliage from the northeastern exposures of Laguna del Hunco (*Carvalho et al., 2013*). The two records of *Todea* in the Tufolitas Laguna del Hunco comprise the only South American fossil or extant record of the genus (extant in Australia, New Guinea, and southern Africa; *Carvalho et al., 2013*).

### Methods

The *Todea* rhizome was studied in serial thin sections using light microscopy. Photographs of whole thin sections were taken using a light box and a Canon EOS 70D Camera fitted with a Canon 60 mm macro lens and were stitched together using Photoshop CC (Adobe, San Jose, California, USA). All other micrographs were taken using a Nikon Coolpix E8800 digital camera mounted on a Nikon Eclipse E400 microscope and processed using Photoshop CC. All specimens and preparations are housed in the collections of the Museo Paleontologico Egidio Feruglio (MPEF-Pb), Trelew, Argentina, under catalog number
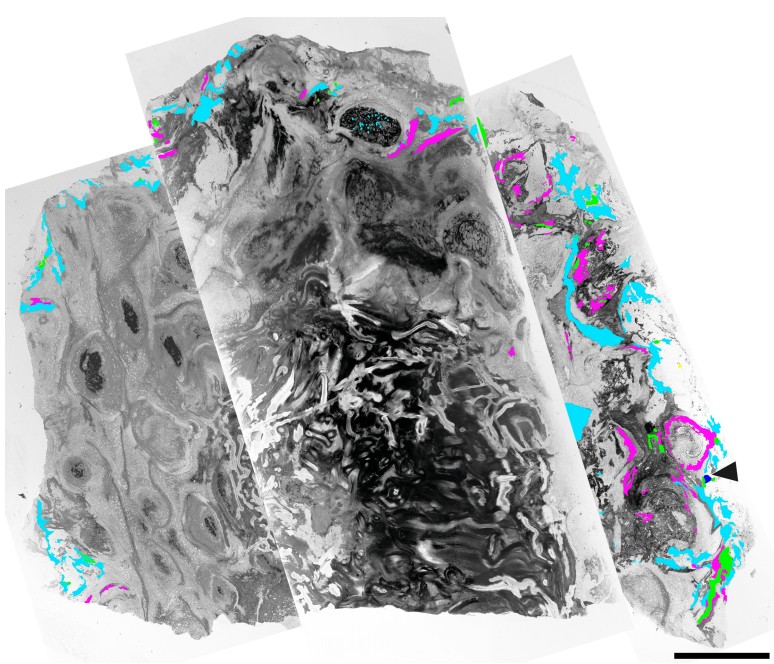

**Figure 1** **Component communities associated with a *Todea* rhizome from the early Eocene of Patagonia, specimen MPEF-Pb 9299.** Composite image of cross section through the rhizome with vascular cylinder at center and numerous leaf traces in a helical pattern. The distribution of associated organisms is mapped in green (for leafy liverworts), light blue (fungi in detritus), purple (coprolites); arrowhead–gymnosperm root. Note denser colonization of the more degraded part of the rhizome at right. Scale bar = 1 cm.

**Table 1** **Distribution of types of organisms colonizing the Patagonian osmundaceous rhizome (data point counts).**

| Fern tissue type | Liverworts | Fungi in detritus | Coprolites | Gymnosperm root | Degraded cell content |
|---|---|---|---|---|---|
| Vascular tissue | 0 | 1 | 8 | 0 | 2 |
| Fiber ring | 1 | 4 | 18 | 0 | 4 |
| Parenchymatous mesophyll | 1 | 77 | 26 | 0 | 1 |
| External to fern tissue | 20 | 88 | 7 | 1 | 0 |
| Total | 22 | 170 | 59 | 1 | 7 |

MPEF-Pb 9299. The distribution of colonizing organisms in fern tissue was quantified by superimposing a 200 μm × 200 μm grid onto the cross section of the rhizome (Fig. 1). At each intersection of the grid, we recorded the type of fern tissue and the presence and types of colonizers. This process yielded 3,820 data points for the entire rhizome cross section. Of these, 252 data points included colonizing organisms (see Table 1).
## RESULTS

### Epiphytic plants

More than 25 small leafy shoots are found, primarily along the outer perimeter on one side of the rhizome, but also between leaf bases elsewhere along the rhizome (Fig. 1). The leafy shoots are up to 3.75 mm long and 230–410 µm in diameter. They exhibit pinnate branching, with branches >200 µm long, diverging at 450–850 µm intervals (Figs. 2B–2C). Stems are 30–180 µm in diameter, comprised entirely of isodiametric to oval parenchyma cells 10–22 µm across and 55–78 µm long (Fig. 2B). Leaves are two-ranked and attached at 10–45 µm intervals (Fig. 2A). Incompletely preserved, they are at least 490 µm long, lack a midrib, and are inserted at 30–40° angles (Figs. 2A–2B). Their thickness (7.5–12.5 µm) indicates that they are probably unistratose.

Additionally, a small root was found in the detritus at the periphery of the rhizome (Fig. 1). The root is 0.7 mm in diameter and has a diarch protostele ca. 100 µm wide (Fig. 2D). Secondary xylem with narrow (10–21 µm) tracheids (Fig. 2D) comprises ca. 75% of the root; some bark is also preserved (Fig. 2D).

### Fungi

Fungal hyphae are abundant in highly degraded parts of the rhizome and in surrounding plant detritus (Figs. 1 and 2L). They fall into two types: (1) smaller, apparently aseptate hyphae 1.5–2.0 µm in diameter; and (2) larger, septate hyphae 3.0–5.0 µm in diameter (Fig. 2L). The latter have septae spaced at 21–22 µm; clamp connections were not observed.

Darkly pigmented, round to oval cerebriform microsclerotia are also abundant in highly degraded parts of the rhizome and in the associated plant detritus (Fig. 1). The microsclerotia are 42–50 µm in diameter and composed of isodiametric to oval cells 4–12 µm in size (Fig. 2F).

Five other types of dispersed fungal reproductive structures are present in detritus associated with the rhizome (Figs. 1 and 2G–2K). These reproductive structures are multicellular (three to eight celled), uniseriate, more-or-less linear spores lacking obvious pores (Figs. 2G–2K); four of the five types are preserved with narrow stalks (Figs. 2G, 2H–2K). Type one consists of stalked, three-celled spores (Fig. 2G) ca. 15 µm long × 6 µm wide. The apical cell is rounded, 4 µm long × 6 µm wide. The middle cell is smaller, rectangular, 3 µm long × 6 µm wide. The basal cell is >10 µm long and 6 µm wide, tapering basally into a stalk 4 µm wide. Type two includes three celled, stalked spores roughly elliptical in shape, ca. 40 µm long and up to 16 µm wide (Fig. 2K). The apical cell is shaped like a truncated ellipse, 18 µm long and 16 µm wide. The subapical cell is rectangular, 11 µm long × 16 µm wide. The basal cell is trapezoidal, 6 µm long and 13 µm wide apically, tapering to 6 µm basally. The stalk tapers to 4 µm. Type three consists of elongate, stalked, 7- or 8-celled spores ca. 35 µm long and 6 µm wide (Fig. 2H). Cells, except for the basal and apical ones, are rectangular, 3–5 µm long × 6 µm wide. The apical cell is rectangular-trapezoidal and slightly smaller, 2 µm long × 5 µm wide. The basal cell is also smaller, rectangular-trapezoidal, 3.5 µm long × 4 µm wide. The stalk tapers to 2 µm. Type four is the most abundant fungal spore, three-celled, stalked and spatulate, ca. 25–30 µm long and 15 µm wide (Fig. 2J). The apical cell is round, 13–17 µm in diameter.

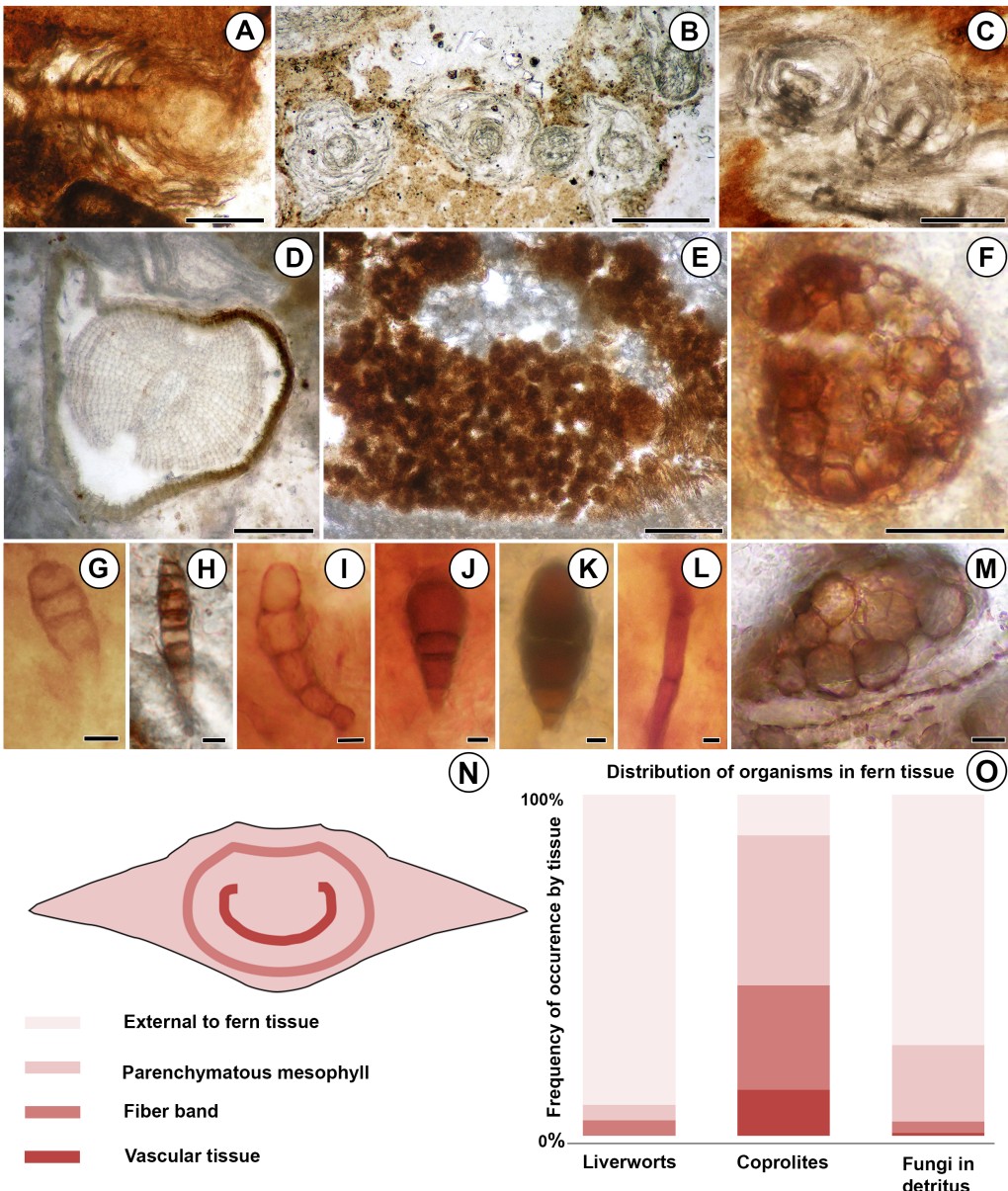

**Figure 2** **Diversity and distribution of organisms associated with the Patagonian *Todea* rhizome.** (A) Longitudinal section of leafy liverwort gametophyte, showing two-ranked arrangement of ecostate leaves. (B) Cross sections of five branches of a leafy liverwort shoot. Note lack of lignified tissues in the stems and thin, ecostate leaves. (C) Oblique longitudinal section of a leafy liverwort stem (bottom) with two diverging branches. (D) Cross section of gymnosperm root with diarch protostele and secondary xylem. (E) Coprolite-filled gallery in osmundaceous leaf base. (F) Cerebriform microsclerotium in detritus around rhizome. (G) Three-celled stalked spores in detritus around rhizome (H) Seven to eight-celled, stalked spores in detritus around rhizome. (I) Five-celled spores in detritus around rhizome. (J) Three-celled, stalked, spatulate spores in detritus around rhizome. (K) Three-celled, stalked, elliptical spores in detritus around rhizome. (L) Septate hyphae in detritus around the rhizome. (M) Circular structures of variable size filling a plant cell, thought to be the result of cell content degradation and aggregation during fossilization. (N) Tissues of an osmundaceous leaf base diagrammed. (O) Frequency of occurrence by tissue (same color coding as N) of each type of organismic remains associated with the Patagonian rhizome. See also Table 1. Scale bars: A–E 250 μm; F 25 μm; G–K, M 5 μm; L 2.5 μm.

The middle cell is small and rectangular, 5–7 μm long × 7–8 μm wide. The basal cell is rectangular-trapezoidal, 8 μm long and 7 μm wide apically, tapering to 5 μm basally. The stalk tapers to 3–4 μm. Type five consists of slightly curved chains of five cells, ca. 38 μm long and 9 μm wide (Fig. 2I). The apical cell is rounded, 8 μm long and 6 μm wide. The sub-apical cell is inflated, 7 μm long × 9 μm wide. The third cell is rectangular-trapezoidal, 7 μm long × 5 μm wide, tapering to 4 μm. The fourth cell is rectangular, 6 μm long × 4 μm wide. The basal cell is square, 4 μm across.

## Coprolites

Small, spherical to ovoid coprolites occur towards the periphery of the rhizome, in excavations of the parenchymatous mesophyll, fiber band, and vascular tissue (Figs. 1 and 2E). Coprolites also occur externally to fern tissues, around highly degraded portions of the rhizome (Fig. 1). They have smooth surfaces and are 23–260 μm in diameter, composed of angular cell-wall fragments and opaque bodies (Fig. 2E).

## Minute spherical structures

The degraded portions of the rhizome have groups of cells that are filled with small, spherical structures 2–6 μm in diameter (Fig. 2M). These structures occur in cells of all tissue types (parenchymatous leaf base mesophyll, fiber band, and vascular tissue; Table 1).

# DISCUSSION

### Taxonomic affinities of epiphytes
#### Small epiphytic plants

The small size of these plants and their lack of conducting tissues indicate that they are bryophytes. The only bryophyte group that combines pinnately branched gametophytes with two-ranked leaves that are unistratose, lack a midrib, and are inserted at wide angles to the stem are leafy liverworts (*Schofield, 1985*). The incomplete preservation of the leaves, which typically provide taxonomically informative characters among liverworts, precludes a narrower systematic placement of these plants.

#### Root

The diarch primary xylem and high proportion of secondary xylem, lacking vessels, are features typical of gymnosperms (*Esau, 1965*).

#### Fungi

The microsclerotia documented here are similar in size and morphology to the cerebriform microsclerotia described in the roots of *Eorhiza arnoldii* Robison et Person from the Eocene of Canada, which were attributed to dark, septate endophytes belonging to a lineage of dematiaceous ascomycetes (*Klymiuk, Taylor & Taylor, 2013*).

Type one spores compare in overall morphology with smaller members of the dispersed fossil spore genus *Diporicellaesporites* Elsik, especially *D. minisculus* Sheffy et Dilcher, but the pores diagnostic of this genus (*Kalgutkar & Jansonius, 2000*) are not observable in the Argentinean material. Type two and four spores are comparable to species of the dispersed spore genus *Brachysporites* Lange et Smith, whose multicellular, usually spatulate spores

resemble conidia of the extant dematiaceous ascomycete genus *Brachysporium* Saccardo (*Kalgutkar & Jansonius, 2000*; *Taylor, Krings & Taylor, 2015*). Within *Brachysporites*, type two spores are most similar to *B. atratus* Kalgutkar, while type four spores are most similar to *B. pyriformis* Lange et Smith (*Kalgutkar & Jansonius, 2000*). Type three spores are comparable to conidia of the extant magnaporthaceous ascomycete *Clasterosporium* Schwein. (*Kalgutkar & Jansonius, 2000*); the extinct *C. eocenicum* Fritel et Viguier is especially similar to our type three spores, although the latter are slightly smaller. Type five spores are comparable to the conidia of the extant pleosporaceous ascomycete *Curvularia* Boedijn based on their size, curved shape, and inflated subapical cell (*Elsik, 1993*).

Overall, the fungal reproductive structures documented on the *Todea* rhizome are comparable with those of several ascomycete lineages, including Dematiaceae (microsclerotia and two of the spore types), Magnaporthaceae, and Pleosporaceae (*Elsik, 1993*; *Kalgutkar & Jansonius, 2000*; *Klymiuk, Taylor & Taylor, 2013*). Septate hyphae lacking clamp connections frequently found in the vicinity of these reproductive structures are consistent with ascomycete affinities. Given that all this fungal material was found in detritus or very degraded plant material (Fig. 1), these fungi, which were a component of the epiphyte community, were probably saprotrophs, rather than parasites or endophytes.

### Coprolites

Based on their size, shape and texture, the excavations and coprolites were probably produced by oribatid mites (*Labandeira, 1998*; *Kellogg & Taylor, 2004*; *McLoughlin & Bomfleur, 2016*).

### Small spherical structures

Their perfectly circular shape, tendency to completely fill cells, and variable size indicate that the small, spherical structures are not microorganisms and are probably not of cellular nature. Instead, they probably represent a type of cell content degradation and aggregation due to the fossilization processes, also seen in the Early Devonian Rhynie chert plants (M. Krings, pers. comm., 2017).

## Distribution patterns

The distribution of colonizing organisms and differential preservation of central vs. peripheral tissues in the *Todea* rhizome indicate that parts of this rhizome were probably alive immediately prior to fossilization. Consistent with this interpretation, vascular tissues near the center of the rhizome are well-preserved and are neither populated by colonizers (Fig. 1) nor degraded (Fig. 3A). However, some of the peripheral leaf bases are highly degraded, surrounded by organic detritus (Fig. 3A), and richly populated with colonizers (Fig. 1). The leaf bases on one side of the rhizome are particularly degraded, and this region is most densely colonized by epiphytic organisms (Fig. 1). Some leaf bases in this region contain coprolite-filled galleries large enough to adversely affect the physiological functions and structural integrity of these leaves (Fig. 3B). Together, these observations indicate that the most degraded and heavily colonized peripheral leaf bases were probably dead at the time of fossilization, whereas the stem and more centrally-located, less degraded leaf bases were probably alive. This distribution of degraded and living tissues is not surprising,

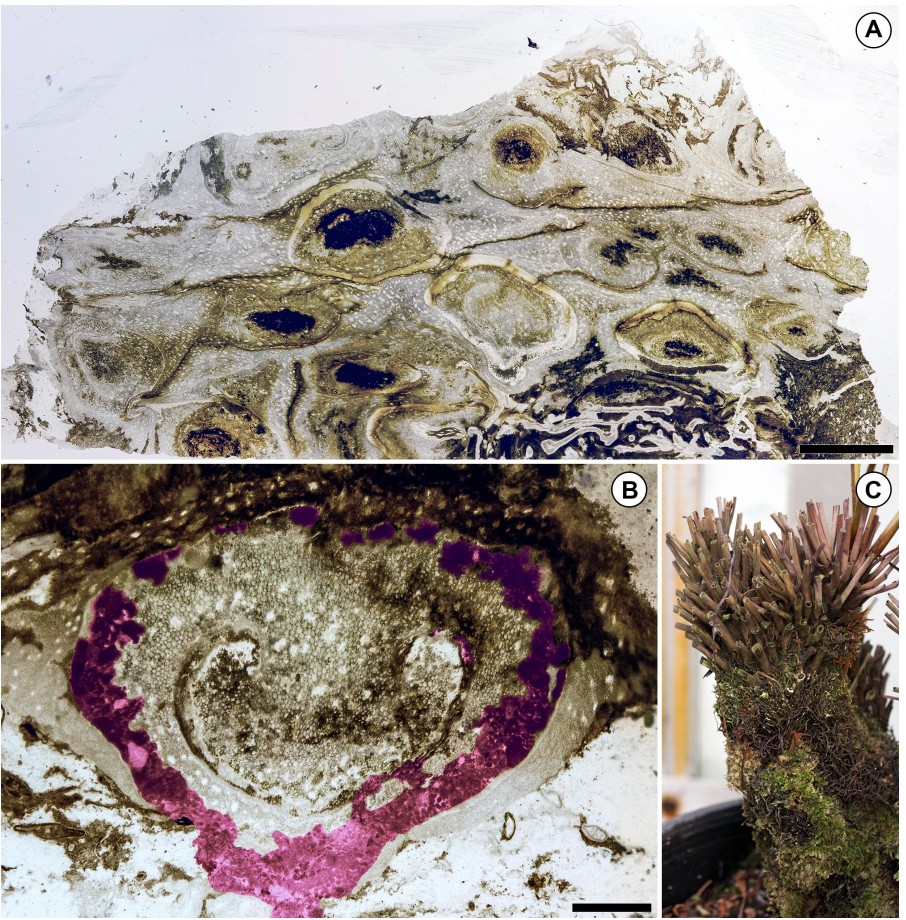

**Figure 3** **Preservation of the Patagonian *Todea* rhizome and densely colonized extant *Osmunda* L. rhizome.** (A) Partial cross section of the rhizome shown in Fig. 1 magnified to show gradient in preservation from periphery to center. (B) Cross section of a degraded leaf base with extensive coprolite-filled galleries. Coprolite-filled galleries are highlighted in purple. (C) Extant *Osmunda regalis* L. rhizome in the Humboldt State University greenhouse showing abundant epiphyte colonization. Scale bars: A 5 mm; B 1 mm.

because the rhizomes of living Osmundaceae are often surrounded by a mantle of degraded and richly colonized dead leaf bases (Fig. 3C).

The fossil *Todea* rhizome exhibits dense colonization by a diverse community of epiphytes, which cover it in several kinds of life (Fig. 1), just like extant osmundaceous rhizomes (Fig. 3C). The distribution and frequency of each kind of organism associated with the rhizome by tissue type (Figs. 2N–2O; Table 1) show that most associated organisms are much more abundant in the more degraded region of the rhizome (Fig. 1). Liverworts and fungi in detritus are found primarily external to the fern tissues, and secondarily within some of the most degraded tissues near the periphery of the rhizome (Figs. 1 and 2N–2O). Coprolites also occur in all types of tissue, as well as externally to some of the most degraded leaf bases, indicating that the arthropods that produced them did not have strong preferences for lignified or parenchymatous tissues (Figs. 1 and 2N–2O).

## Comparison with the Jurassic community associated with *Osmundastrum pulchellum*

A community of organisms similar to that of the *Todea* rhizome has been described associated with an exceptionally well preserved *Osmundastrum pulchellum* Bomfleur, G. Grimm et McLoughlin rhizome from the Jurassic of Sweden (*Bomfleur, McLoughlin & Vajda, 2014*; *McLoughlin & Bomfleur, 2016*; *Bomfleur, Grimm & McLoughlin, 2017*). Like the Patagonian community, this Jurassic community included fungi, oribatid mites (coprolites) and epiphytic plants. However, in the case of the *Osmundastrum pulchellum* community, the epiphytic plants are exclusively vascular (lycopsids and ferns), which may indicate that bryophytes had not yet evolved the epiphytic habit during the Jurassic. Additionally, fungal material is more abundant and diverse in the Patagonian rhizome, probably correlated with higher proportions of decomposed tissues. The richness of the communities associated with both of these osmundaceous rhizomes suggests that other permineralized fern rhizomes are likely to yield diverse communities of associated organisms.

## Significance of the leafy liverwort fossils

Given the sparse fossil record of liverworts (*Oostendorp, 1987*; *Tomescu, 2016*; *Heinrichs et al., 2018*; *Tomescu et al., 2018*), it is not surprising that these tiny epiphytes represent the first report of fossil leafy liverworts from South America. *Marchantites hallei* Lundblad (*Lundblad, 1955*), a thalloid liverwort from the Early Cretaceous of Argentina, is the only other unequivocal fossil liverwort known from South America (*Oostendorp, 1987*; *Tomescu et al., 2018*). However, three additional compression fossils from the Carboniferous of Bolivia and the Mesozoic of Argentina that lack cellular detail have also been compared with thalloid liverworts (*Jain & Delevoryas, 1967*; *Cardoso & Iannuzzi, 2004*; *Coturel & Savoretti, 2018*).

The liverworts described here are also the first epiphytic leafy gametophytes documented in the fossil record outside of amber deposits [see *Grolle & Meister (2004)*, *Frahm (2010)* and *Heinrichs et al. (2018)* for reviews of bryophytes preserved in amber]. All other non-amber fossils of epiphytic bryophytes are moss protonemata described on angiosperm leaves (*Mägdefrau, 1956*; *Selkirk, 1974*; *Barclay et al., 2013*).

## Importance of understanding epiphyte communities in fossil biotas

Today, rich epiphyte communities are found on plants in several biomes, including boreal forests (*McCune, 1993*), tropical lowland rainforests (*Cornelissen & Ter Steege, 1989*), and tropical montane forests (*Wolf, 1993a*; *Wolf, 1993b*). In these ecosystems, epiphyte communities provide critical ecological services, including soil production within forest canopies (*Enloe, Graham & Sillett, 2006*), increased nutrient input (*Coxson & Nadkarni, 1995*), and increased water storage (*Pócs, 1980*; *Veneklaas & Van Ek, 1990*; *Pypker, Unsworth & Bond, 2006*). Thus, epiphyte communities significantly influence the ecology of their host plants. The makeup of epiphyte communities is diverse in the modern biota and co-varies dramatically with forest composition. In boreal forests of Pacific Northwest North America, lichens and mosses may dominate (*McCune, 1993*), whereas bromeliads

and orchids may dominate epiphyte communities in a mid-elevation rainforest of central Mexico (*Hietz & Hietz-Siefert, 1995*). The evolution of the deeply divergent epiphyte communities in these biomes is critical for understanding the ecology and evolutionary history of the forests themselves. However, our understanding of epiphyte community composition in the distant past is poor and based on a small number of examples (e.g., *Rothwell, 1991*; *Rössler, 2000*; *McLoughlin & Bomfleur, 2016*).

Knowledge of complex epiphyte communities in fossil plant assemblages, including detailed insight into component communities like those of the Patagonian rhizome, is needed to paint a more complete picture of life in the geologic past and contextualize the evolution of epiphyte communities. Past epiphyte communities probably provided the same kinds of ecosystem services as those of the modern biota because these functions are independent of epiphyte community structure. Thus, attempts to understand the ecology of these fossil biotas without epiphytes ignore a critical component of the ecosystem. Similarly, an understanding of ancient communities is required to place their modern counterparts in an evolutionary framework (*Gerhold et al., 2018*).

## Osmundaceous rhizomes as a model system for understanding epiphyte community structure

In principle, an appropriate host organism could provide a model system to track epiphytes through geologic time, and from this we could determine when certain groups evolved the epiphytic habit (e.g., modern leptosporangiate ferns, lycophytes, bryophytes, angiosperms) and how epiphyte community structure varied between different forest types in the distant past. A model system would also allow for rigorous testing of hypotheses on the evolution of epiphytic leafy liverworts (*Feldberg et al., 2014*), lycophytes, and filicalean ferns (*Lovis, 1977*; *Schneider et al., 2004*). Such a host organism would need to meet four requirements: (1) a morphology that facilitates colonization by epiphytes; (2) morphological stasis over a wide stratigraphic range; (3) widespread geographic distribution; and (4) abundance in the fossil record.

Permineralized osmundaceous rhizomes satisfy these requirements. (1) The leaf bases found at the periphery of osmundaceous rhizomes have a rich microtopography that facilitates colonization by epiphytes. Indeed, living osmundaceous fern rhizomes are often covered in epiphytes (Fig. 3C). (2) Osmundaceous ferns have been in morphological stasis since the Permian (>250 million years; *Miller, 1971*; *Bomfleur, Grimm & McLoughlin, 2017*). (3) Osmundaceae were widespread geographically in the geologic past (*Miller, 1971*; *Bomfleur, Grimm & McLoughlin, 2017*). (4) Osmundaceous ferns arguably have the richest fossil record of any living fern lineage (*Arnold, 1964*; *Miller, 1971*; *Tidwell & Ash, 1994*; *Bomfleur, Grimm & McLoughlin, 2017*). Additionally, living osmundaceous ferns allow direct comparison of the fossil epiphyte communities to their extant counterparts.

To date, permineralized osmundaceous rhizomes have yielded evidence for plant interactions with invertebrates (*Schopf, 1978*; *Tidwell & Clifford, 1995*; *McLoughlin & Bomfleur, 2016*) and fungi (*Kidston & Gwynne-Vaughan, 1907*; *Gould, 1970*; *McLoughlin & Bomfleur, 2016*). These occurrences confirm that osmundaceous rhizomes were important hosts for epiphyte communities in ancient ecosystems, just as they are today. Because

permineralized osmundaceous rhizomes represent a well-populated epiphyte niche that has remained unchanged over time and space and have a rich fossil record reaching into the Permian, they provide a paleoecological model system for exploring epiphyte community structure and evolution.

## CONCLUSIONS

The complex community of organisms associated with an early Eocene osmundaceous fern rhizome from Patagonia allows a glimpse of a tiny, ancient epiphyte community centered around a single host plant. This community is surprisingly diverse and includes the first fossilized leafy liverworts in South America, also marking the only fossil record of leafy bryophyte epiphytes known outside of amber deposits; several types of fungal hyphae and spores; microsclerotia with possible affinities in several ascomycete families; and coprolites produced by oribatid mites.

Understanding of complex epiphyte communities in fossil plant assemblages is needed to accurately understand ecological networks within extinct ecosystems and to unearth the evolutionary history of extant epiphytes. In the fossil record, complex epiphyte communities on plants have only been characterized in uncommon instances (e.g., the petrified forest of Chemnitz, Germany; *Rössler, 2000*). In contrast to these rare occurrences, fossil osmundaceous ferns and the communities they may harbor are widespread geographically and stratigraphically (*Stewart & Rothwell, 1993*; *Taylor, Taylor & Krings, 2009*). Osmundaceous ferns can be traced back into the Paleozoic (*Miller, 1967*; *Miller, 1971*; *Stewart & Rothwell, 1993*; *Taylor, Taylor & Krings, 2009*; *Bomfleur, Grimm & McLoughlin, 2017*), a time when terrestrial ecological networks were becoming canalized (*DiMichele et al., 1992*). Furthermore, osmundaceous morphology has been in stasis since the Paleozoic (*Eames, 1936*; *Stewart & Rothwell, 1993*; *Bomfleur, McLoughlin & Vajda, 2014*). As such, osmundaceous rhizomes represent a distinctive ecological niche, unchanged since the Permian, and act as hosts for epiphyte communities comprised of plants, animals, and fungi. These communities can also be studied in the modern biota and can be sampled systematically and consistently across all occurrences, fossil and extant. Because of the scale of the organisms (centimeter to decimeter), entire *in situ* tiny epiphyte communities can be studied directly in fossil specimens. Thus, osmundaceous rhizomes represent an ecological niche that has remained unchanged over time and provide a paleoecological model system for exploring epiphyte community structure across geologic time and space. Given the diversity of epiphytes associated with osmundaceous rhizomes from the Huitrera Formation and the Jurassic of Sweden (*McLoughlin & Bomfleur, 2016*), as well as the abundance of osmundaceous rhizomes in the fossil record (*Miller, 1967*; *Miller, 1971*; *Taylor, Taylor & Krings, 2009*; *Bomfleur, Grimm & McLoughlin, 2017*), we predict that continued exploration will reveal additional aspects of the biotic networks centered around these important repositories of ecological data from the geologic past.

## ACKNOWLEDGEMENTS

The authors thank Carla J. Harper, (University of Kansas, Lawrence, Kansas) and Terry Henkel (Humboldt State University, Arcata, California) for help identifying fungal material associated with the fern rhizome, Michael Krings (Ludwig-Maximilians-Universität *München*, Germany) for help determining the origin of the degraded cell contents, Benjamin Bomfleur (Universität Münster, Germany) for identifying the host fern, Eduardo Ruigomez (Museo Paleontológico Egidio Feruglio, Trelew, Argentina) for facilitating work in the MPEF collections, and Russell Bryan for assistance rendering Fig. 2N. The authors are also grateful to Gar Rothwell (Oregon State University, Corvallis, Oregon), Benjamin Bomfleur (Universität Münster, Germany), and one anonymous reviewer whose comments significantly improved the manuscript.

### Funding

This research was supported by U.S. National Science Foundation grants IIA-1322504 (Alexandru M.F. Tomescu), DEB-1556666, and EAR 1925755 (Peter Wilf), the American Philosophical Society (Alexandru M.F. Tomescu), and by a Paleontological Society G. Arthur Cooper Award, a Botanical Society of America graduate research award, the Alistair and Judith McCrone Graduate Fellowship (Humboldt State University), and a U.S. National Science Foundation Graduate Research Fellowship (No. 1546593) to Alexander C. Bippus. The funders had no role in study design, data collection and analysis, decision to publish, or preparation of the manuscript.

### Grant Disclosures

The following grant information was disclosed by the authors:
U.S. National Science Foundation: IIA-1322504, DEB-1556666, EAR 1925755.
American Philosophical Society.
Paleontological Society G. Arthur Cooper Award.
Botanical Society of America graduate research award.
Alistair and Judith McCrone Graduate Fellowship (Humboldt State University).
U.S. National Science Foundation Graduate Research Fellowship: 1546593.

### Competing Interests

Peter Wilf is an Academic Editor for PeerJ.

### Author Contributions

- Alexander C. Bippus conceived and designed the experiments, performed the experiments, analyzed the data, contributed reagents/materials/analysis tools, prepared figures and/or tables, authored or reviewed drafts of the paper, and approved the final draft.
- Ignacio H. Escapa, Peter Wilf and Alexandru M.F. Tomescu analyzed the data, contributed reagents/materials/analysis tools, authored or reviewed drafts of the paper, and approved the final draft.

## Data Availability

The raw specimen counts are published in Table 1. The fossil material is permanently housed in the Museo Paleontologico Egidio Feruglio (MPEF-Pb), Trelew, Argentina, under catalog number MPEF-Pb 9299.

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
