# Peer review of "Fossil fern rhizomes as a model system for exploring epiphyte community structure across geologic time: evidence from Patagonia"

_PeerJ, doi:10.7717/peerj.8244_

## Round 0.1 · original submission · Major Revisions

Overall, I consider the topic and the research presented in this manuscript to be within the scope of PeerJ and of a quality that would contribute positively to the journal. However, at this time, the paper appears to need revision, possibly of a major sort.

The two reviews make quite different recommendations but both raise, implicity or explicitly, several issues that might be addressed further in this paper. One is the matter of the longitudinal temporal distribution of anatomically preserved osmudaceous rhizomes, including extant forms. Reviewer 1 requests considerably more information about the plant, including greater specification and description of its morphology, and a taxonomic identification. These seem reasonable requests to me. The reviewer also comments on the growth dynamics of fern rhizomes, which may be alive at one end and dead and decaying some distance back from the apex - this almost certainly needs to be addressed, especially in light of comments made by Reviewer 2.

Reviewer 2, like Reviewer 1, finds merit in aspects of the paper. However, this reviewer questions the certainty of the assertion that the fungi were part of the living fern-bryophyte microecosystem. Also questioned is the concept of a "model system" - basically a model of what? One would use a model system to understand some sort of evolutionary or ecological phenomena, or theoretical construct, which would be illuminated by the system in a particularly important way that could be extended to other, less well understood instances.
Reviewer 2 also notes that aspects of the organization of the manuscript are in need of attention. Reviewer 1 has made a small number of editorial suggestions in the annotated manuscript, but these point to some more general issues with the writing, which could use another careful going-over.

Finally, I considered a third review, but thought it best, at this point, given the substantial matters raised by the two reviewers, to allow the authors to consider these suggestions, respond if needed, and/or make changes and incorporate suggestions.

·

Basic reporting

See general comments to the editor

Experimental design

See general comments to the editor

Validity of the findings

See general comments to the editor

Additional comments

Review of Fossil fern rhizomes as a model system for biotic interactions across geologic time: Evidence from Patagonia (#21919)

This study introduces and documents an exciting approach for characterizing epiphytic ecosystems and separating them from detrital ecosystems in a relatively consistent environment over an extensive segment of the Phanerozoic fossil record. The paper is well organized and clearly written in excellent English. With the exception of one figure, it is well illustrated with the appropriate photos and one table of excellent quality. Identifications and characterizations of the different organisms appear to be sound. They are appropriately documented with photographic evidence and citations to the appropriate literature. Overall, the literature is appropriate and up to date.

I find this to be an important paper for demonstrating the potential of the fossil record of plants. It extends further into the realm of extinct ecosystems the types of studies that can be accomplished with fossils. The paper will be of interest to a broad spectrum of paleobotanists, paleobiologists, and paleoecologists.

Figure 1A needs to be improved in quality and possibly increased in size to show better the features of the osmundaceous stem, and of the various mircoenvironments present within a fern trunk. It would be helpful if the authors could offer a genetic identification of the osmundaceous fern. In addition, a few edits have been offered on the accompanying PDF of the submitted manuscript. Otherwise, this is an excellent contribution that I recommend be accepted with minor revisions. A few minor suggestions follow:

Lines 124-133 – Fossil osmundaceous fern trunks with anatomical preservation are well known, and well characterized to genus. What is this? If possible, please identify to genus and figure features that will document more than merely the ordinal identification of this fern. None of the characters listed on lines 130-132 (except phyllotaxis) are clearly illustrated in Figure 1A.

Line 233 – Fern rhizomes may be growing at the apex and dead below. In fact, most tree ferns have stems that are dead and rotting at the base, but are kept upright and nurished by the stem-borne adventitious roots. As a result, whether the fern was alive or dead at the time of fossilization isn’t as important as whether this section of the stem was alive or dead at the time of fossilization. Please clarify this in the text.

GWR

Reviewer 2 ·

Basic reporting

This study presents a new record of an Eocene permineralized osmundaceous rhizome that contains abundant evidence of bryophytes growing on the rhizome surface and several different fungal remains associated with the peripheral, partially decayed regions and tissue systems of the rhizome. The discovery is interpreted as an interaction system, and it is concluded that osmundaceous stems in general may represent paleoecological model systems providing insights into biotic interactions across geologic time.

The ms meets the basic professional requirements for article structure, language use, and data sharing.

Experimental design

No comments

Validity of the findings

The findings are potentially worthy of being considered for publication in PeerJ, but I feel that the ms as stands is insufficient. My gravest concern relates to the fact that the word ‘interactions’ is used to circumscribe and qualify the assemblage of organisms seen in this fossil. But what the fossil actually shows are ill-preserved bryophytes that grew on the surface of the rhizome, as well as several different types of fungal spores, hyphae, and microsclerotia that were produced by some fungi, which perhaps lived in close association with the fern (within the tissues or on the surface), but could equally well be fungi that occurred in the substrate surrounding the fern, with propagules accidentally transported into the degrading periphery of the rhizome. Speaking of ‘interactions’ therefore is over-interpreting the data! The bryophytes are associated with the fern – ok. Conversely, the fungal hyphae, spores, and sclerotia, based on the evidence provided, just occur and co-occur in the partially degraded periphery of the rhizome. As to whether the fungi that produced the hyphae, spores, and sclerotia actually interacted (i.e. entered into some form of specifiable relationship) with the fern cannot be determined based on the material at hand. Part of the problem results from the lack of information regarding the number of specimens of the individual fungal remains. (1) Has each of the types of fungal remains only been found once or a few times, or do all these different remains occur frequently in the rhizome periphery? (2) If the latter is accurate, then, where exactly do they occur? Randomly distributed or in specific areas of the rhizome, perhaps in areas into which they could not accidentally become washed and that would indicate spore production within host tissue? (3) What exactly does the mapping of the different remains on a single cross section of the rhizome tell us? To provide any meaningful insight into what you call a “model system” (e.g., with regard to infection and distribution pathways of the fungi; tissue preferences etc.), a quantitative analysis and mapping of the whole series of sections is needed.

Permineralized osmundaceous rhizomes are referred to as a “paleoecological model system” that can be used to explore plant biotic interactions across geologic time. As stated above, there is no easy way to establish based on the data at hand that interaction actually occurred between either of the organisms found on and in this rhizome. Moreover, I do not quite understand what exactly the ‘model’ character is? I am thinking, if you take three rhizomes, one from the Permian, the second from the Triassic, and the third from the Neogene, analyse all three for microbial content, and plot the results, what exactly is this going to tell you? You will likely see differences in the number and quality of remains, along perhaps some similarities. So what? As stands, it reads as if you view the rhizome-bryophyte-microorganism assemblage as a super organism with a very long evolutionary history, and thus that changes in the assemblage of organism reflect something that has to do with paleoecology. But is it really that easy?
In conclusion, I feel that much more work is needed to evaluate in full the many different and interesting aspects preserved in this rhizome. I strongly feel that the original research results presently are too slight to vindicate publication. It will be necessary to provide a more extensive description and photographic documentation of the bryophytes and microbial remains, especially with regard to their spatial distribution. Finally, the technical flaws of the writing (e.g., amalgamation of descriptive and discussion items in the Results section) need to be removed.

Additional comments

See above

---

## Round 0.2 · Minor Revisions

I believe, based on the expert review, that this paper is very close to acceptable for PeerJ. Please take note of the reviewer's comments in the annotated manuscript. These are not extensive, but they do raise some important points about the taxonomy. There also are a number of excellent editorial suggestions, which will improve what is already a well written manuscript.

It should be noted that the authors responded very well to the earlier reviews, and made important changes in the earlier submitted manuscript.

It is unlikely that any further review will be necessary, but the authors should respond to the comments in the annotated manuscript - please list these comments/responses consecutively in a cover letter.

·

Basic reporting

see attached pdf

Experimental design

see attached pdf

Validity of the findings

This is my first review of this manuscript altogether.
I base my suggestion on an evaluation of the manuscript in its present form in light of the comments and suggestions accompanying the previous editorial decision and of the revision notes of the authors accompanying this revised submission.
In general, I understand this project to be more of a pilot study. Having worked on this particular fossil myself, I can agree to the authors’ decision to refrain from a more in-depth and more quantitative analysis, as reviewer 2 has (justifiably) requested earlier. The preservation of the fossil is simply too incomplete and too imperfect to provide sufficiently adequate material for a more detailed analysis. As we state in our description of the fossil, which has only just been published, “[…]preservation of the fossil material presented here is imperfect: most of the stem parenchyma appears to have been degraded and replaced by silica, and many of the more robust, lignified or sclerified structural elements in the stem appear crushed or variably displaced, which makes precise interpretation of many of the structural features commonly used for species delimitation in Osmundaceae (e.g., numbers of leaf traces in cortical regions or of xylem strands per stem cross section) difficult.” (Bomfleur and Escapa, 2019). This would apply similarly to studying the inventory and distribution of epiphytic organisms.
Still, even though the authors present largely qualitative data and an only informal systematic treatment of the epiphytic organisms, I do see great merit in publication of this revised manuscript version pretty much as is, if only to provide a survey of the evidence of biotic interactions in the fossil, and to provide an impetus for further study of similar material from different localities and different stratigraphic ages.
In this context and with this scope, I find this manuscript to present novel and interesting findings that contribute significantly to our understanding of such complex component communities in the geological past. The manuscript is well-written and well-prepared; the structure has been revised, and results and discussion more clearly separated during revision following previous editorial suggestions. Some key issues that the previous reviewers put forward, such as the request for a more detailed description and taxonomic placement of the host fern, have become obsolete with the separate formal systematic treatment of the fossil (Bomfleur and Escapa, 2019).
All in all, I come to suggest publication following a minor final revision; I only have a few remaining issues that I think should be addressed (see comments and suggestions in the attached annotated pdf file).
Hoping I could be of assistance, I remain
Yours sincerely,

Benjamin Bomfleur

Additional comments

Dear Alex Bippus, dear co-authors,

I have reviewed (only) the revised of your manuscript and come to suggest publication following just a minor reviison of a few remaining issues.

Best wishes,

Benni

---

## Round 0.3 · accepted · Accept

Thanks for the prompt, thorough response, and for (again) careful consideration of the reviewer's comments and suggestions. This is a very nice paper that should prove of interest to a wide audience of neo and paleo botanists.